# Impact of Drought on Soil Microbial Communities

**DOI:** 10.3390/microorganisms13071625

**Published:** 2025-07-10

**Authors:** Sujani De Silva, Lithma Kariyawasam Hetti Gamage, Vesh R. Thapa

**Affiliations:** Department of Agronomy and Horticulture, University of Nebraska-Lincoln, Lincoln, NE 68583, USA; mdesilva2@huskers.unl.edu (S.D.S.); lkariyawasamhettig2@huskers.unl.edu (L.K.H.G.)

**Keywords:** climate change, drought stress, soil microbes

## Abstract

Drought, an increasingly prevalent climate stressor due to global warming, profoundly impacts agricultural systems, particularly the soil microbiome. Soil microorganisms are crucial for nutrient cycling, plant health, and ecosystem stability; however, drought-induced changes disrupt microbial community structure, function, and interactions with plants. This review synthesizes current knowledge on the effects of drought on soil microbiomes, with a focus on microbial diversity, resilience, and functional shifts in agricultural contexts. It highlights key microbial mechanisms underpinning plant drought tolerance, including symbioses with plant growth-promoting bacteria and fungi. Furthermore, it addresses knowledge gaps in the long-term effects of repeated drought events, microbial adaptations, and plant–soil feedback mechanisms. By advancing our understanding of drought–microbiome dynamics, this review aims to inform sustainable agricultural practices and resilience strategies to mitigate the adverse impacts of drought on crop productivity and ecosystem health.

## 1. Drought Stress and Climate Change

Drought is a natural disaster that occurs when there is a prolonged period with insufficient precipitation, resulting in a shortage of water resources. The impacts of drought on agriculture are significant, and one of the most critical consequences is the alteration of soil habitat, which can have a significant impact on the microbial structure and function. Microbes play a vital role in maintaining the health and productivity of soil ecosystems, and drought conditions can negatively affect their populations and diversity. Changes in microbial populations can alter the availability of nutrients, impact the growth of plants, and ultimately affect crop yields. Therefore, it is essential to understand how drought affects microbiomes in agriculture and how we can manage these impacts to improve crop productivity and sustainability. In this context, studying the effects of drought on soil microbiomes can provide valuable insights into the ecological mechanisms underlying drought response in agricultural systems, which can help us develop effective strategies for managing these impacts.

Climate change is increasing the frequency, duration, and severity of droughts. One of the reasons that is acknowledged for drought severity and occurrence is global warming. The global surface temperature was found to have increased by 0.99 and 1.09 °C over two 19-year periods, from 2001 and 2011 to 2020, respectively [1]. Berg & Sheffield [2] explained that higher temperatures and changes in precipitation patterns contribute to increased evapotranspiration rates and higher average aridity scores for many landscapes. The water limitation condition shares pronounced impacts in semi-arid and arid regions across the world including South America, Asia (India, Sri Lanka), and Africa [3,4]. The global increase in temperature and its associated effects are major reasons why predicting and evaluating climate change scenarios is crucial, especially for identifying areas most at risk of drought stress during the dry season.

Drought stress significantly alters the physio-chemical properties of soil, directly and indirectly influencing soil microbial communities. During the drier season, when water availability decreases, drought becomes a dominant stressor that directly and indirectly affects microbial communities and their habitats. It influences microbiome resiliency under drought stress and leaves legacies that contribute to crop growth, health, yield, plant–microbe interactions, and overall ecosystem services [5,6,7]. Under water-limited conditions, microbial abundance declines, community structures become destabilized, and microbial activity—including enzymatic processes. Those carried out by oxidoreductases, dehydrogenases, and phosphatases—is reduced [7].

Similarly, high temperatures can affect the growth and activity of soil microbes, leading to changes in the soil microbial community structure and function [8]. Drought directly affects plant morphology, anatomy, physiology, and biochemistry. Its effect on plants can also be observed by changes at the transcriptomic and metabolomic levels. However, in plants, it can be mitigated by rhizosphere microbial communities, especially by plant growth-promoting bacteria (PGPB) and fungi (PGPF) that adapt their structural and functional compositions to water scarcity [7].

If severe drought stress occurs over a long period of time, this will need further research to investigate the continuous worsening effects that follow. Additionally, it remains uncertain whether post-drought conditions can be fully restored to their original state [5,6,7]. As such, understanding the effects of drought on soil habitat is essential for managing and conserving soil resources in the face of climate change.

Overall, studying drought is crucial for understanding the impacts of climate change on the agriculture microbiome. By elucidating the mechanisms by which drought affects soil microbial communities and their interactions with plants, we can develop strategies to mitigate the adverse effects of drought on crop productivity and soil health.

While there has been prior work characterizing changes in microbiome structure under drought conditions, the effects of prolonged stress (press disturbance) and increasing frequency of drought events (repeated pulse disturbances) are not well understood. Evaluating changes in community composition and diversity following disturbance events is essential for identifying the characteristics of resilient microbiomes [7,9,10,11,12], which may be managed under real-world conditions to ameliorate drought stress in cropping systems. Additionally, the link between changes in microbiome structure and microbial community function is poorly understood, particularly in response to drought stress in agricultural systems across diverse regions and with differing host crops. Elucidating patterns of microbiome function and not only microbial community structure will be essential for understanding how drought events may mediate plant–soil feedback to shape crop health and soil fertility. This will be particularly important for addressing the role of climate change in shaping spatiotemporal variability in plant–microbe interactions that may destabilize beneficial associations or heighten the risk of pathogenicity and disease [13]. Similarly, there is a need to understand how drought stress impacts the abundance, diversity, and functional roles of specific microbial groups that support crop health, including plant growth-promoting bacteria and mycorrhizae. Illuminating how microbe–microbe interactions are mediated by drought stress can provide unique insight into how beneficial functional traits may be promoted in agroecosystems [7,11].

The objective of this review is to highlight the latest literature on drought stress and its effects on soil habitat and microbiome structure and function, identify key knowledge gaps, and propose future research directions. Addressing these gaps and proposing future direction is critical in developing adaptive practices to mitigate drought impacts and sustain agroecosystem productivity in the face of climate change.

### Materials and Methods

This review synthesizes findings from peer-reviewed research and review papers published between 2000 and 2023. Relevant sources were identified using databases on Google Scholar. The selected studies encompass a wide range of geographical regions and cropping systems, offering a comprehensive analysis of microbial responses to drought. Particular emphasis was placed on experimental and observational studies that explore the connections between soil habitat, microbial diversity, community structure, and functional processes under drought conditions.

## 2. Effect of Drought on Soil Habitat

Drought can have a significant impact on the soil habitat, which encompasses the physical, chemical, and biological environments in which soil organisms live and interact. Soil habitat is essential for supporting plant growth and ecosystem functioning, as it provides a source of nutrients, water, and other resources necessary for life. During a drought, soil water content decreases, which can lead to changes in soil structure, nutrient availability, and microbial activity. These changes can negatively impact the soil habitat, making it less hospitable for many soil organisms [2,14,15]. While some legume crops like common beans, beach beans, and chickpeas are known for their drought tolerance [16,17], drought stress can still disrupt the soil habitat. It does so not only by directly altering soil moisture and temperature but also indirectly by affecting soil carbon and nitrogen availability, shifting soil pH, and influencing plant growth dynamics. These combined effects further contribute to the degradation of soil conditions, making it more difficult for soil organisms to thrive. Changes in soil temperature during drought conditions can affect soil organic matter (SOM) decomposition and increase the release of carbon dioxide. Also, during this process, additional mineral N, mostly in the form of nitrate, will be released into the soil system. Recognizing how drought shapes soil environments is important for understanding how this stressor may drive changes in microbiome structure and function.

### 2.1. Soil Moisture

Droughts typically occur when there is a prolonged period of low precipitation or when there is high evapotranspiration due to hot and dry weather conditions [18]. As a result, the soil becomes increasingly dry, and soil moisture is depleted, which can lead to changes in the structure and composition of soils [18]. As soil moisture decreases, the soil water potential becomes more negative, creating a gradient that pulls water away from plant roots [2]. This water is lost to the atmosphere through evaporation and transpiration, and is also retained in small soil micropores, making it less available to plants [2]. Reducing soil moisture can indeed lead to soil compaction [2,19]. When soil moisture levels decrease, the soil particles are more likely to come into close contact with each other, reducing the amount of pore space between them [2,19]. This can result in a reduction in soil volume and an increase in soil density, which is referred to as soil compaction [2,19]. In addition, when the soil becomes drier, it can become more difficult for roots to penetrate and grow, which can further contribute to soil compaction [20]. This is because the roots are not able to break up the soil as effectively when it is dry, and so the soil particles remain tightly packed together. The soil can become more compact, reducing soil porosity and oxygen availability, which can negatively impact the growth and activity of aerobic soil microbes [21]. When soil moisture is too high, it can reduce the amount of oxygen available to soil organisms, leading to anaerobic conditions that can be harmful or even lethal to many types of soil organisms [22]. Excess moisture can also increase the risk of fungal and bacterial diseases, which can further harm soil organisms [22]. Conversely, when soil moisture is too low, soil organisms may struggle to obtain the water they need for survival, growth, and reproduction. This can lead to dehydration and reduced activity, and in severe cases, mortality [22]. Further, when the soil is dry, the forces that hold water in the soil, namely capillary forces and the attraction between water and soil particles, become stronger [2]. This makes it more difficult for plants to access water [22], which can negatively feed back to hinder microbial growth and activity in soils and the rhizosphere [22]. Soil moisture content is the primary driver of changes in soil habitat following drought stress and can alter nutrient availability across plant–soil compartments.

### 2.2. Soil Carbon Availability

Drought can have a significant impact on soil carbon storage, as it affects the balance between carbon inputs and outputs in the soil ecosystem.

During drought periods, plant productivity is often reduced. Deng et al. [23] discovered that drought significantly decreased soil organic carbon content by 3.3%, primarily due to an 8.7% decline in plant litter input and a 13% reduction in litter decomposition across all three ecosystem types—forests, shrubs, and grasslands—worldwide. Globally, significant reductions in C of leaves, shoots, roots, and bulk soil were seen under drought conditions. During drought periods, plants decrease the carbon inputs into the soil through photosynthesis and root exudation. Furthermore, soil carbon storage also decreases during drought periods through increasing the rate of microbial/soil respiration, which releases carbon dioxide from the soil into the atmosphere. This decline may have been caused by lowered stomatal conductance and decreased activity of vital photosynthesis-related enzymes, including ribulose-1,5-bisphosphate carboxylase/oxygenase. These elements probably decreased net primary production (NPP) and plant C concentrations, both of which affected the amount of organic matter added to the soil [14,15].

In addition, drought can increase the rate of microbial/soil respiration, which releases carbon dioxide from the soil into the atmosphere, further reducing soil carbon storage [24]. Based on previous meta-analysis data which includes 107 studies, drought stress significantly decreased soil C by an average of 12.8% and ranged between 18.4% and 7.2% [24]. Measured extractable organic carbon was significantly reduced by drought stress in 0–15 cm, 15–30 cm, and 30–45 cm depths the whole year from November 2018 to August 2019 in the Poplar plantation field in China [24]. The effect of drought on soil respiration, however, was found to be dependent on the severity and duration of the drought. Drought-induced changes in soil moisture and temperature are the primary drivers of the observed changes in soil organic carbon (SOC) concentrations and turnover rates [25]. Specifically, drought reduces soil moisture, which limits microbial activity and reduces plant growth, resulting in lower inputs of organic matter into the soil. Drought also increases soil temperature, which accelerates microbial decomposition of organic matter and reduces the stability of soil aggregates, further contributing to the loss of SOC [26].

### 2.3. Soil Nitrogen Storage

Drought can have a significant impact on the nitrogen pool of soil (Figure 1). Nitrogen is an essential nutrient for plant growth and is present in the soil in various forms, including organic nitrogen, ammonium (NH_4_^+^), and nitrate (NO_3_^−^). Drought conditions can reduce microbial activity in the soil, which can decrease the rate of organic matter decomposition and nitrogen mineralization. Drought stress can also affect plant growth and reduce their ability to take up nitrogen from the soil. This can lead to a buildup of available nitrogen in the soil, as plants are not using it for growth [23].

Drought increased mineral N (NH_4_^+^ and NO_3_^−^) content (31%) due to decreased mineral N uptake, lower litter and root productions, and reduced NO_3_^−^ leaching during drought. Plant growth is restricted under decreased water availability across various ecosystem types (forest, shrubs, and grassland), leading to less litter and root production [23]. Furthermore, fewer roots inevitably decrease the uptake of mineral N [27]. Fewer roots absorbed less available N (e.g., NH_4_^+^ and NO_3_^−^) from the soil [28]. Drought reduced N mineralization rate (5.7%) and nitrification rate (13.8%) and thus left the total N unchanged across all the three ecosystem types (forest, shrubs, and grassland) in the world. The reason for that is that drought limited microbial activities, leading to lower N mineralization and nitrification rates [23]. Grossiord et al. [29] found that summer drought in a California annual grassland increased exchangeable NH_4_^+^, which is explained by reduced N uptake by plants, as the grasses had died, and by microbial death and the subsequent mineralization of microbial N.

Also, drought reduces the soil C:N ratio due to disproportionate losses of carbon relative to nitrogen, largely driven by microbial mortality [14,15,23]. Under drought stress, microbial activity often declines due to limited soil moisture, slowing the decomposition of organic matter. However, prolonged drought can also cause microbial death, particularly among organisms responsible for carbon cycling. The release of organic compounds from dead microbes can accelerate the mineralization of carbon, while nitrogen may remain more stable or become immobilized. This imbalance results in a net loss of carbon compared to nitrogen, ultimately decreasing the soil C:N ratio.

**Figure 1 microorganisms-13-01625-f001:**
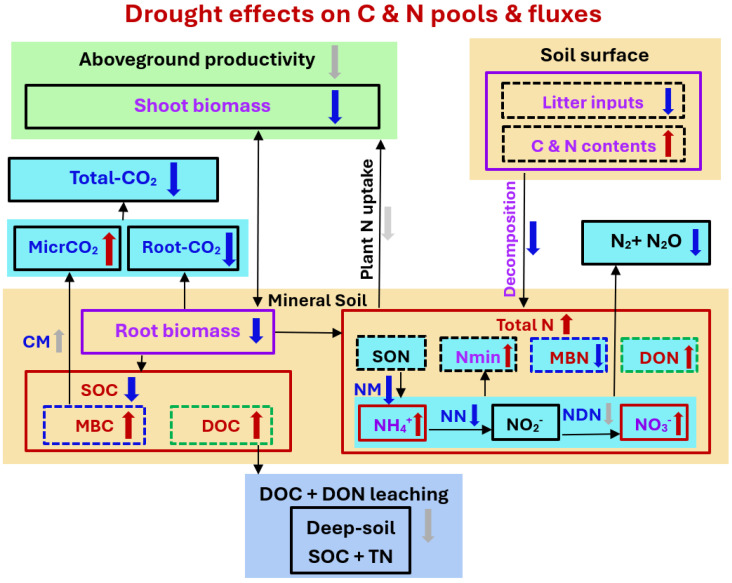
Conceptual framework illustrates the impacts of drought on soil carbon (C) and nitrogen (N) pools and fluxes. Red arrows (“↑”) indicate increases, blue arrows (“↓”) indicate decreases, and gray arrows represent inferences or processes from previous studies. The figure shows observed drought-induced changes in pools such as shoot and root biomass, soil organic C (SOC), microbial biomass C (MBC), dissolved organic C (DOC), microbial biomass N (MBN), dissolved organic N (DON), mineral N species (NH_4_^+^, NO_2_^−^, NO_3_^−^), and associated fluxes including total CO_2_ efflux, root and microbial respiration (Root-CO_2_, MicrCO_2_), and N emissions (N_2_ + N_2_O) [23]. Black outlined boxes denote pools or fluxes excluded in previous meta-analyses. The figure also includes processes like plant N uptake, soil N mineralization (NM), nitrification (NN), denitrification (NDN), and leaching of DOC and DON [23,30]. The diagram strictly reflects the observed directions and magnitude changes under drought compared to non-drought conditions.

Soil N pools, plant N inputs from litter and roots into soils, soil N turnover processes—including mineralization, ammonification, nitrification, and denitrification—N leaching, and N uptake by plants and microorganisms are all influenced by drought.

### 2.4. Soil pH

Drought can have a significant impact on soil pH, although the exact effects can vary depending on the soil type and the severity and duration of the drought. Drought can lead to salt accumulation in the soil through evapotranspiration, which typically increases soil pH and makes the soil more alkaline, but in fertilized soils, nutrient uptake may be lower locally [31].

In dry soils, pH is high, due to drought stress in forest and Tundra of Western Sayan Mountains, Southern Siberia [32]. When the soil is unable to hold onto water, these organic acids are not retained in the soil and are instead leached out. This causes a decrease in the concentration of organic acids in the soil, which in turn reduces the acidity of the soil. Song et al. [33] found that decreases in pH under water stress in the rhizosphere for drought-tolerant crops still produce the exudates and increase nutrient availability. DesRochers et al. [34] found that drought treatment only slightly reduced growth and rarely interacted with N source or soil pH.

## 3. Effect of Drought on Soil Microbial Structure

Drought can directly affect microbes through desiccation and resource limitation as substrate diffusion is reduced at low levels of soil moisture [35]. Drought experiments have reported decreases in microbial biomass and activity, a reduction in carbon and nitrogen mineralization [36], and the accumulation of solutes—amino acids (in bacteria) and polyols (in fungi)—which help prevent dehydration but are energetically expensive [37]. Drought may also have indirect effects through interactions with plants, as plants can have species-specific effects on rhizosphere microbiota mediated by rhizodeposits [38]. Severe drought conditions modify the microbial community structure, biomass, and activity in soils. Understanding how different fractions of the microbiome respond to drought stress is crucial for understanding how abiotic stress may shape crop–microbe interactions and influence the resiliency of agroecosystems. Evaluating how microbial community structure shifts immediately following drought events and tracking the time it takes for communities to recover is important for improving our understanding of microbiome resilience.

### 3.1. Drought as an Ecological Disturbance

As a disturbance event in ecological systems, drought effects can be categorized as either a press or pulse disturbance, depending on their duration, frequency, and ecological impact [39]. A press disturbance is defined as a prolonged and continuous stressor that typically lasts longer than one growing season, exceeding the system’s natural recovery capacity. Such disturbances may lead to chronic stress, cumulative ecosystem impacts, and persistent shifts in community composition and function. In contrast, a pulse disturbance refers to a short-term event, typically lasting less than three months, that causes a temporary disruption in ecosystem processes but allows for recovery once the stressor is removed. This classification provides a more transparent framework for interpreting the ecological consequences of drought based on its temporal characteristics [39,40]. Although drought is often viewed as a press disturbance due to its association with prolonged water deficits, it can also be classified as a pulse disturbance under certain conditions, when it is brief (e.g., lasting weeks to a few months), infrequent, and occurs in resilient ecosystems capable of rapid recovery (e.g., semi-arid grasslands adapted to seasonal dry spells). In such cases, drought does not lead to irreversible changes, and the system resumes normal functioning with the return of moisture. Therefore, whether drought acts as a press or pulse depends on its duration relative to ecosystem recovery capacity: “prolonged” in a press context implies a disturbance lasting beyond the system’s ability to recover, while a pulse is short enough for the system to bounce back without lasting damage [39,40,41]. Press disturbance can result in a prolonged water shortage, which can lead to selective mortality of certain plants and microorganisms that are unable to tolerate the prolonged water scarcity [6]. The effects of drought on selective mortality and demographic and/or adaptive rescue depend on the severity, duration, and frequency of the drought [5]. However, some plant species may be better adapted to survive under drought conditions and may reproduce more successfully during periods of drought. This can lead to changes in the population structure of the ecosystem, with an increase in drought-tolerant plant species. For example, deep-rooted perennial grasses, succulents, or drought-adapted shrubs can become more dominant [7]. Additionally, some plant species may adapt to drought over time through a process known as adaptive rescue. This involves the evolution of traits that allow species to cope with prolonged drought conditions, including the ability to store water, deeper roots, or the ability to reduce water loss through transpiration [8]. Over time, such adaptations can enhance the ecosystem’s resilience to future drought events [12].

Understanding how different fractions of the microbiome respond to drought stress is important for understanding how abiotic stress influences crop–microbe interactions and impacts agroecosystem resiliency. For example, drought can cause significant shifts in microbial community structure, favoring drought-tolerant microbial taxa that can maintain essential functions under water-limited conditions [11]. Evaluating how microbial community structure shifts immediately following drought events and tracking the amount of time it takes for communities to recover is important for improving our understanding of microbiome rescue [7].

### 3.2. Impact of Soil Moisture on Microbial Activity

Soil moisture is a primary aspect that regulates the survival and activity of microorganisms. Drought conditions have detrimental effects on soil biochemical properties since microorganisms that are not adapted to high water tension may not survive under these unfavorable conditions [42]. The soil subjected to severe water stress may be responsible for the low microbial biomass due to mechanisms of reduced diffusion of soluble substrates and/or reduced microbial mobility, resulting in limited access to substrates [30]. Not only physiology but also the soil microbial community structure is affected by the changes in water status [7,43]. Several types of microorganisms are affected differently by changes in water potential due to drought [30,43]. Fungi were not affected by the drought compared to the bacteria [44].

In some cases, changes in soil moisture may have a minor impact on the composition of the microbial community. Dissimilar types of microorganisms are differentially affected by varying amounts of water potential [45]. For example, Gram-negative type bacteria are thought to be more sensitive to dramatic changes in water potential, while fungi have been implicated as more tolerant of low water availability [46]. Many management practices, along with plant and soil factors, serve as important mediators of soil microbial community structure [45].

### 3.3. Variability of Drought Effects on Soil Microbial Communities

The effect of drought stress on microbial communities can vary in several types of soils or under soil modifications [47]. Severe drought conditions modify the microbial community structure in soils [32,48]. However, their effect on the microbial structure is more significant in soils with low organic matter content. Under controlled drought conditions, significant shifts were observed in the relative abundances of microbial phyla in both sandy and loamy soils, characterized by low organic matter content [49,50]. Among six phyla—*Actinobacteria*, *Bacteroidetes*, *Firmicutes*, *Planctomycetes*, *Proteobacteria*, and *Verrucomicrobia*—that together accounted for more than 95% of the total bacterial abundance, *Actinobacteria* (especially the genera *Gaiella* and *Nocardioides*) were the most prevalent in the analyzed samples. In contrast, the relative abundance of *Proteobacteria*, *Corticotrophs*, and *Verrucomicrobia* decreased significantly [49]. In a study by Bastida et al. [51], soils later in the growing season, which were affected by drought, had higher abundances of *Proteobacteria*, *Actinobacteria*, *Chloroflexi*, and *Nitrospirae* than non-drought-affected soils that were dominated by *Acidobacteria* early in the growing season. There were also changes in the fungal community [52].

Carbon and nitrogen availability is more important for microbial growth. Microbial phyla, including *Verrucomicrobia*, *Proteobacteria*, and *Acidobacteria*, which are sensitive to nitrogen ratios, tend to decline during drought because drought induced reductions in labile carbon and nitrogen entering the rhizosphere [50,53,54].

Bacteria may be more sensitive than fungi due to their lifestyle strategies. Bacteria are less motile and more reliant on water availability to create channels and connect spaces, facilitating access to nutrients or exchanging genetic material [53]. Gram-positive bacterial lineages are more drought-resistant than Gram-negative lineages, due to their thicker cell walls [37].

### 3.4. Impact of Drought on Microbial Biomass and Recovery

The impacts of drought on soil microbial biomass can be explained through direct effects mediated by the lower soil moisture in drought-affected plots, as observed here, as well as through plant-mediated effects. Bastida et al. [51] indicated in their study that drought reduced belowground carbon allocation by plants, which coincides with the lower survival of phosphorus. Moreover, reduced soil moisture and pore connectivity in drought-exhausted plots would have limited the transfer of organic matter from plants to the soil, as well as the formation of organo-mineral complexes. The combined analysis of the total and active microbial community composition allowed the ecological adaptations of phyla to climate change to be deciphered [49,51]. A reduction in carbon allocation in soils subjected to drought can limit the growth and activity of *Proteobacteria*, including copiotrophic taxa that are strongly related to ecosystem multifunctionality [51]. *Actinobacteria* is one of the most enriched bacterial taxa in drought-treated soils across a range of environments [55] and in drought-treated rhizospheres for several plant species [54]. *Actinobacteria* can be enriched due to their spore-forming ability, which allows them to enter a stable and dormant state during periods of environmental stress, thereby enabling them to persist under drought conditions [54]. Naylor et al. [35] investigated how *Actinobacteria* become enriched within plant roots under drought conditions, revealing that the plant host strongly influences their enrichment patterns. The researchers found that Actinobacteria were more significantly enriched in the root endosphere compared to the surrounding soil. Furthermore, the study provided evidence that the composition and enrichment of different microbial taxa under drought conditions are correlated with the evolutionary histories of their plant hosts, highlighting the importance of host identity in shaping drought-responsive root microbiomes.

### 3.5. Drought Effects on Fungal and Bacterial Communities

Drought can decrease bacterial biomass, and the magnitude of this effect can be intensified under nutrient-poor soils. Kohler et al. [56] reported that under drought stress, the total PLFAs for Gram-positive bacteria were significantly lower, with the decline being more pronounced in soils that did not receive organic amendments compared to those that did. Similarly, the Gram-negative bacteria in the wheat rhizosphere were significantly lower under drought conditions, and this effect was more pronounced in rhizospheric communities from Luvisol soils compared to Chenozem soils [57]. Drought stress is contributing to decreases in bacterial biomass and persistence. Drought stress has been shown to reduce bacterial biomass and persistence in soils, primarily due to limited water availability and disruption of microbial activity [49]. However, soils with higher organic matter content can buffer these adverse effects, as organic matter helps retain moisture and provides a more stable environment for microbial communities. As a result, bacterial populations in organic-rich soils are generally more resilient to drought compared to those in soils with low organic matter [50,58]. Drought duration significantly increased the fungal-to-bacterial ratio, suggesting that fungi are more resistant and functionally important than bacteria under water stress conditions. [59]. This is attributed to the chitinous cell walls of fungi, which are thought to increase their resistance to environmental fluctuations, particularly water stress [60]. Additionally, fungal hyphal growth enables them to cross small areas of dry soil. Fungi are thought to be more resilient and/or resistant to drought conditions than bacteria. For example, the formation of extensive hyphal networks enables fungi to access water from distant soil microhabitats, while the production of resilient spores allows for long-term survival under dry conditions through dormancy. This functional stability of fungal communities under drought has been supported by studies, notably the study by de Vries et al. [61], which highlights the persistence of fungal networks during soil desiccation events. Evidence suggests that yeasts among fungi may exhibit higher drought tolerance, as they are more prevalent in extreme environments and reproduce through budding, a stress-tolerant reproductive strategy [62]. Analysis of microbial diversity through co-occurrence networks highlights distinct patterns among bacterial communities compared to fungal communities [61]. Bacterial networks contain central or hub operational taxonomic units (OTUs) sensitive to drought conditions and serve as indicators for recovery after drought stress. In contrast, fungal communities exhibit more stable networks under drought conditions, showing less sensitivity to drought stress and higher resiliency [61,63].

### 3.6. Microbial Effects on Plants Under Drought Conditions

The total fungal community and active microbiome were impacted by drought. This is because the fungal community was more sensitive to the lowered soil moisture than the bacterial community [64]. Different mechanisms can explain the distinct sensitivity of total bacterial and fungal communities to drought. For instance, Kaisermann et al. [64] suggested that fungal community composition depends on non-extreme moisture conditions. As an example, Bastida et al. [51] proved that drought negatively affected the activity of *Eurotiales* and *Hypocreales*, and the reduction in their activity was related to lower mycorrhizal fungi (MF) abundance. They also indicated that some fungal populations could coexist by occupying different moisture niches [64].

Microbial effects on plants can be neutral or positive, although soil feedback effects are generally negative, depending on the different microbial groups involved [7]. In fact, the responses of plant species to environmental changes, such as drought, may also be influenced by soil microorganisms [7,65]. For example, mycorrhizal fungi may benefit water-stressed plants by increasing access to soil water, improving plant hydraulics and gas exchange, whereas fungal pathogens may exacerbate plant vulnerability to drought [65,66].

## 4. Effect of Drought on Microbiome Functions

### 4.1. Enzymatic Activity in Soil

Soil enzymes, produced by animals, plants, and microorganisms, play a crucial role in soil health and nutrient cycling. Microorganisms are the primary producers of extracellular enzymes, including β-glucosidase, hydrolases, urease, phosphatase, cellulase, amylase, and chitinase [36,67,68]. These enzymes are essential for the biodegradation of organic compounds in soil and serve as indicators of microbial activity, responding to environmental changes, including drought. Enzymatic activity in soil can reflect shifts in microbial community behavior, often providing insight into the soil’s biological response to water stress [36,69,70].

The impact of drought on soil enzymes varies and is not always straightforward. A reduction in soil moisture typically leads to a decrease in microbial biomass, which in turn reduces soil enzyme activity. Studies have shown that enzyme activities, namely laccase, glucosidase, phosphatase, and chitinase, decrease with low soil moisture levels [71,72]. However, the specific responses of these enzymes can vary depending on the soil type, crop species, and farming practices. Enzyme responses vary contextually: glucosidase increases in sandy soils under moderate drought but decreases in clay soils under severe stress [57].

Research on the effect of drought on enzymatic activity in soils has shown that different enzymes respond to water scarcity in unique ways. For example, in Mediterranean evergreen oak forests, urease, protease, and β-glucosidase activities were significantly reduced under drought conditions, with the greatest reductions occurring in autumn and at deeper soil depths [73]. Other enzymes, like phenol oxidases, exhibited different patterns, with some activities increasing in response to seasonal variations in moisture. These findings highlight the complexity of enzymatic responses to drought, which is influenced by both the specific enzyme involved and the surrounding environmental conditions [73,74].

Drought influences soil enzyme activities by altering microbial populations and their metabolic functions [48,49]. Enzymes crucial for nutrient cycling often exhibit decreased activity under dry conditions, although some enzymes may show adaptive increases in response to specific stressors [48,49]. Understanding these changes is important for assessing soil health and the resilience of microbial communities under water-limited conditions [48,49].

### 4.2. Nutrient Availability and Potential Nutrient Cycling

Severe drought conditions can significantly compromise nutrient availability in the soil, as demonstrated in the study by Hueso et al. [36]. They found that in drought-stressed soils, soil dehydrogenase activity increased (accounting for 34% of the total variability), while other enzyme activities decreased, alongside high basal respiration and water-soluble carbon content. The increase in water-soluble carbon may be attributed to an increase in carbohydrate production (52% of the total variability) as fewer soil microbes produce biological polymers under low-water conditions. Drought may also cause the death of sensitive microorganisms unable to thrive under such harsh conditions [36]. This, in turn, leads to the release of substrates from dead cells into their surroundings, providing accessible nutrients to the drought-resistant microorganisms or survivors [36]. Additionally, the decreased metabolic activity of the soil microbial community under drought conditions leads to a reduction in the mineralization of carbon, nitrogen, and phosphorus, thereby affecting their biological cycles and pathways [36]. This suggests that prolonged periods of drought stress result in a decline of overall microbial biomass, reducing functional diversity in the soil [12,48].

Drought strongly affects soil nitrogen cycling by inhibiting nitrification [49]. The nitrification potential (NP), a highly sensitive parameter reflecting soil microbial response to environmental factors including moisture content, was significantly reduced during drought. Siebielec et al. [49] showed that after one month of drought, NP activity decreased by 70% to 80% in loamy and sandy soils, respectively. These results suggest that sandy soils with low organic matter content are more susceptible to drought stress than loamy soils. Interestingly, the application of compost to sandy soils slightly alleviated the negative effects of drought on soil nitrifying bacteria [49]. The impact on nitrification underscores how drought affects not only microbial survival but also significantly alters nutrient cycling, with a particular emphasis on nitrogen transformation in the soil.

Microbiome structural changes under drought conditions are closely linked to shifts in microbial community function. Characterizing these shifts, including changes in nutrient cycling, the abundance of key genes or functional traits, and metabolic signatures (e.g., phytohormones), is critical for understanding how microbial communities modify their environment and influence crop growth during drought. Under drought conditions, the microbial community becomes less active, leading to a reduced release of labile carbon and nitrogen, which is essential for the soil ecosystem [10]. As bacterial growth and decomposition slow, some microbes enter dormancy or produce spores that can grow when water is available again [10]. Additionally, a study on the inoculation of sugarcane with *Bacillus subtilis* during drought revealed that the inoculated plants had increased nitrogen (N), phosphorus (P), calcium (Ca), magnesium (Mg), and sulfur (S) content, along with improved chlorophyll levels and higher photosynthesis rates compared to non-inoculated plants under water restrictions [75]. This synergy between *B. subtilis* and the host plant enhanced nutrient uptake and improved drought tolerance by stimulating the expression of stress-response genes, phytohormones, and related metabolites [75]. This process also enhanced phosphorus solubilization through acidification, chelation, and organic acid production, which helped mitigate drought stress in the plant and supported improved photosynthetic activity.

### 4.3. Microbial Metabolite Production and Gene Expression Shifts

Drought triggers microbiomes to biosynthesize secondary metabolites and sugars, enhancing pathways for osmolyte production [76]. Drying soil has been shown to cause shifts in microbial functional genes, leading to increased transcripts associated with nucleotide metabolism, adenylyl transferase, polyribonucleotide nucleotransferase, DNA-directed DNA polymerase, and DNA topoisomerase [76]. For example, transcription levels for trehalose gene production, including those encoding trehalose phosphatase, increase significantly under soil drying conditions [76]. Most taxa exhibit higher levels of gene expression during drought [76]. Additionally, microbial metabolite profiles shift, showing an increase in sugars (simple and carboxyl acids) and sugar alcohols. Alongside these changes, carbon metabolites also increase, promoting carbon fixation [76].

### 4.4. Impact of Arbuscular Mycorrhizal Fungi (AMF) and Plant Growth-Promoting Rhizobacteria (PGPR) on Drought-Tolerant Crops

Silva et al. [77] studied the inoculation of soil with arbuscular mycorrhizal fungi (AMF), plant growth-promoting rhizobacteria (PGPR) *Bacillus subtilis*, or both. In soils inoculated with AMF and subsequently planted, severe drought increased the abundance of AMF hyphae, while moderate drought increased the formation of AMF vesicles, both of which contributed to greater root colonization. Phosphatase activity was changed during severe and moderate drought. Acidic phosphatase increased only in AMF-only samples and decreased in PGPR-only and AMF + PGPR samples. Alkaline phosphatase activity increased with PGPR-only samples, although the increase was only slight. But this phosphatase decreased drastically with AMF-only and AMF + PGPR samples. All inoculants showed increased P uptake and nutrient acquisition, resulting from the synergistic plant–microbe relationships.

AMF are recruited by plants through phytohormones and signaling molecules, like strigolactone, which are released into the rhizosphere [78]. AMF will then grow into the plant roots and can modulate them to become highly plastic like AMF, which enhances water uptake and helps minimize water loss [78]. The reduction in water lost during drought stress enables the plant to continue growing [78]. AMF’s increased hyphal network and glomalin secretion assists with water and nutrient uptake. Extra-radical hyphae increase the plant’s root architecture, root hydraulic conductivity, and photosynthetic rate (helps regulate stomatal conductance) [78].

### 4.5. Role of Plant Growth-Promoting Rhizobacteria (PGPR) in Mitigating Drought Stress

Plant growth-promoting rhizobacteria (PGPR) form symbiotic relationships with plant hosts, which can help mitigate drought stress in plants. PGPR contribute to bacterial nitrogen fixation (BNF), enhance nitrate reductase activity, synthesize phytohormones (auxins, cytokinins, gibberellins, ethylene), solubilize phosphate, and improve systematic resistance to drought and tolerance to abiotic stress [79]. Inoculating legumes with both *Azospirillum* and *Rhizobia* promotes early nodulation, increases BNF rates, and supports plant growth and yield [79]. This symbiotic relationship offers protection against abiotic and biotic stresses as *Azospirillum* produces phytohormones that extend the plant’s root system and improve root architecture [79]. Enhanced root architecture boosts the plant’s ability to absorb nutrients and water while increasing enzyme expression necessary for detoxifying reactive oxygen species (ROS) [79]. Additionally, these microbes produce secondary metabolites that influence defense hormones, signal symbiosis, and promote growth in both plants and microbes [79]. For instance, *Rhizobia*’s nod factor establishes effective symbiosis with legumes [79].

## 5. Future Needs and Approaches

Based on findings, drought is one of the most significant climate hazards affecting agriculture worldwide. Its impact on crop yields can be severe, leading to food insecurity and economic losses for farmers. The effects of drought vary across different climates worldwide. However, it is important to understand the variation in drought stress impacts on plant–soil relations within different climatic zones for future needs. However, information about it is still limited and not widely available in all climatic zones. As such, it is important to gather more specific data, including metatranscriptomic analyses, to understand the structure and function of crop-associated microbiomes in regions projected to face heightened drought stress. For instance, in the United States, drought can cause significant yield losses for maize, soybeans, and wheat, which are the main crops grown in the country [80,81]. In Africa, drought can reduce the yield of staple crops for maize, sorghum, and millet, which are crucial for food security in the region [82,83]. In Asia, specifically in tropical countries like India and China, drought is linked to a decline in the yield of the most essential crops like rice, maize, and soybean [84,85,86]. In scale of global study for maize and wheat, Daryanto et al. [87] explained that the global yield production is reduced by 39.3% for maize and 20.6% for wheat at 40% water reduction. The sensitivity to drought was found to be higher in corn, regardless of whether it was grown in dryland or non-dryland regions. Meanwhile, wheat cultivation in the dryland region is more susceptible to the effects of drought

Based on the background and effects of drought stress, which disrupts the ecosystem stability and yield, it is crucial to understand the processes involved in maintaining the long-term health and productivity of ecosystems. Drought can lead to a decrease in soil moisture, affecting microbial communities essential for nutrient cycling and plant growth. Additionally, drought can alter the structure and diversity of microbial communities, resulting in cascading effects on the entire ecosystem. Subsequently, drought influences plant–soil feedback mechanisms by altering soil microbial communities, nutrient availability, and root growth patterns. These changes can vary depending on the crop species and regional environmental conditions. In semi-arid regions, drought can reduce beneficial soil microbes for sensitive crops like wheat and maize, limiting nutrient and water uptake, while drought-tolerant crops like sorghum maintain more stable microbial relationships. Regional soil properties also influence how drought affects soil and plant interactions, making the impact of drought on plant–soil feedback crop- and region-specific. Lastly, to address this issue, it is essential to develop effective agronomic strategies to mitigate the negative impact of drought [6,88].

Currently, while there is growing research on the influence of drought on microbiome structure, the impact of this stress on microbiome resiliency is poorly characterized. Studies that consider drought stress as pulse or press disturbances and aim to elucidate the responses of microbiomes across temporal scales are needed to understand better how abiotic stress can alter plant–soil–microbe interactions. This is particularly important in the context of plant–soil feedback, where the frequency, duration, and intensity of drought events contribute to changes in plant growth, soil characteristics, and microbial community dynamics which can feed back to shape crop growth and microbial ecology across spatiotemporal scales [61,89]. Plant–soil feedback plays a crucial role in determining the ecosystem’s resilience to drought stress. Drought stress can alter the composition of soil microbial communities, resulting in changes in soil structure, nutrient cycling, and the decomposition of organic matter. These changes, in turn, can affect plant growth and development [6]. One of the areas for improvement is to understand the specific mechanisms of each crop, and further research is needed to elucidate the molecular and biochemical processes involved in plant–soil feedback (PSF) under drought stress. Information on the PSF on different soil types and various agricultural management practices is still limited, and a deeper study is needed to understand whether the PSF undergoes similar complex and multiple mechanisms or not.

One of the key mechanisms of PSF involves modulating plant exudates, which can alter the composition and function of the soil microbial community [90]. Under drought stress, plants may release specific exudates that promote the growth of drought-tolerant microbial communities or inhibit the growth of pathogenic microbes, thereby enhancing plant growth and stress tolerance. Studying PSFs under drought conditions is crucial for enhancing our understanding of ecosystem resilience. Ecosystem resilience refers to an ecosystem’s ability to recover from disturbances and maintain its structure and function. Understanding the mechanisms driving PSFs under drought stress can inform strategies for enhancing ecosystem resilience and productivity under water-limited conditions, which are expected to become more frequent and severe with climate change [61,89].

In response to the changing environment caused by water stress conditions, recent studies have identified specific microbial taxa that are adapted to drought conditions [11,12]. Despite these promising results, a significant knowledge gap remains regarding the mechanisms underlying microbial adaptation to drought conditions. Further research is needed to determine the specific genetic and biochemical adaptations that enable these microbes to survive and thrive in water-limited environments. Understanding these mechanisms could help inform the development of strategies to enhance the resilience of soil microbial communities and maintain soil health during drought stress. Another critical knowledge gap is the need to study the long-term effects of drought on soil microbial communities. Most studies on the topic have focused on short-term drought stress; however, in reality, drought conditions can persist for years or even decades in some regions. The long-term effects of drought on soil microbial communities, and how these effects may impact plant growth and productivity, remain poorly understood. Addressing this knowledge gap is essential for developing effective strategies to mitigate the impacts of drought on agricultural productivity and food security.

As the continuous effects of global warming persist and correspond to drought stress, it is important to identify future directions and next steps to address this issue. Previous research suggests that breeding and integrating drought-tolerant varieties will be a key solution to address drought stress issues. Drought-tolerant crops have special coping mechanisms that enable them to enhance their growth and development, even under scarce-water conditions. Moreover, some agronomic practices have been implemented to reduce the negative effects of drought on soil, some of which include the application of additional organic matter like by-products (sewage clogs, compost, biofertilizer) that help to increase soil C and soil enzyme activities, which will later be beneficial for improving water holding capacity and aggregate stability in the soil. Organic matter that contains rhizosphere microbial communities, like plant growth-promoting bacteria and fungi (PGPB and PGPF), will be more helpful in enhancing water content in the soil. Biomolecular technology and DNA sequencing of soil samples from the rhizosphere of healthy crops can be used to identify any potential microorganisms that help under drought conditions [91,92].

## 6. Conclusions

This review provides an integrative synthesis of the current knowledge on the effects of drought on soil microbial communities in agricultural systems. It emphasizes that drought is intensified by global climate change. It significantly alters soil physicochemical properties, disrupts microbial community structure and function, and weakens plant–microbe interactions. These changes can lead to reduced crop productivity and long-term degradation of soil health. The review further highlights that fungi often exhibit greater drought resilience than bacteria due to structural and functional traits like spore formation and hyphal networks. While short-term responses have been relatively well characterized, substantial knowledge gaps remain regarding the long-term impacts of press disturbances, microbial adaptation mechanisms, and the dynamics of plant–soil feedback under recurring drought events. Addressing these gaps is crucial for developing microbiome-informed strategies that utilize plant growth-promoting microorganisms and organic amendments to enhance drought resilience in cropping systems. Future research should focus on linking microbial community structure with function, unraveling the molecular basis of drought tolerance, and tailoring region- and crop-specific approaches to ensure sustainable food production in a changing climate.

## Data Availability

No new data were created or analyzed in this study.

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
