# Peer review of "Impact of Drought on Soil Microbial Communities"

_microorganisms, 2025, doi:10.3390/microorganisms13071625_

Round 1
Reviewer 1 Report (Previous Reviewer 1)
Comments and Suggestions for Authors
Manuscript Number: microorganisms-3705362
Title: Impact of Drought on Soil Microbial Communities
This paper reviews the effects of drought on soil microbial communities, especially in agricultural systems. This topic is of great significance in the context of global climate change, because the increase in drought frequency and intensity poses a serious threat to agricultural production and ecosystem health. The direct effects of drought on microbial diversity, community structure and function were discussed. The interaction between microorganisms and plants, especially the role of plant growth promoting bacteria and fungi, was also analyzed in depth. This provides a new perspective for understanding the mechanism of plant drought tolerance. Knowledge gaps in current research were identified, especially regarding the long-term effects of repeated drought events, microbial adaptation, and plant-soil feedback mechanisms. Filling these knowledge gaps will help develop sustainable agricultural practices and drought strategies.
The discussion in some parts of the article is lengthy. Repeating the same research results may affect the fluency of reading and the concentration of information. The citation of some data and conclusions in the article is not completely aligned with the original literature, which may cause readers to question the accuracy of the data.
Concrete contents
1 grammar
Original : Drought is natural disaster, revised : Drought is * * a * * natural disaster.
Original article : Microbes play vital role, corrected : Microbes play * * a * * vital role.
Original : Higher temperature... was found to increase, revised : Higher temperatures... * * were * * found to increase.
The original text : Berg & Sheffield [ 2 ] explained that higher temperatures... * * are contributing * *, correction : unified tense → * * contribute * * ( scientific facts use the simple present tense ).
Original article : gram-negative bacteria, corrected : * * Gram-negative * * bacteria ( Microbiological terms are hyphenated and capitalized ).
'This review rought stress ' → should be 'This review discusses drought stress ' or 'This review focuses on drought stress '.。
"it is crucial for understanding the process for maintaining the health and productivity of ecosystems in the long term" → A clearer expression should be "it is crucial to understand the processes involved in maintaining the health and productivity of ecosystems over the long term."
"Subsequently, the drought also affects the plant-soil feedback where each crop in certain region" → revise "Subsequently, drought also affects plant-soil feedback, with each crop in certain regions being impacted."
"is critical for developing adaptive practices" → revise "is critical in developing adaptive practices"。
"drought stress that disturb" → revise "drought stress that disturbs" 或 "drought stress, which disturbs"。
"drought can lead to a decrease in soil moisture, which can affect the microbial communities that are essential for nutrient cycling and plant growth" → "Drought can lead to a decrease in soil moisture, affecting microbial communities essential for nutrient cycling and plant growth."
2. Content of logic
1 ) Effects of drought on soil habitat ( Section 2 )
Question : Contradictions in soil pH discussion
Original text ( Page 6 ) :
Drought can lead to buildup of salt... making soil more alkaline [ 20 ]. Also, soil pH can be decreased in fertilized agricultural land. '
Contradictory point : Undistinguished conditions ( salinization vs.fertilized farmland ).
Recommendations for amendment :
Drought typically increases soil pH via salt accumulation, but in fertilized soils, nutrient uptake may be lower locally. '
2)Microbial community response ( Section 3 )
Problem : The conclusion of fungal drought tolerance lacks data support.
Original article ( Page 8 ) :
'Fungi are thought to be more resilient... than bacteria. '
Logical defects : Unexplained mechanisms ( such as hyphal network / spore dormancy ).
Suggestions for improvement : Citing specific studies ( such as de Vries et al., 2018 ) to illustrate the stability of fungal networks.
3 ) Microbial Functions ( Section 4 )
Problem : Contradictory expression of changes in enzyme activity
Original text ( Page 10 ) :
'Drought stress can enhance glucosidase activity... but in other cases causes a decrease. '
Logical loophole : Conditional differences ( e.g., soil type / drought intensity ) are not clarified.
Recommendations for amendment :
'Enzyme responses vary contextually : glucosidase increases in sandy soils under moderate drought but decreases in clay soils under severe stress [ 58 ].
4 ) Missing chart reference
Question : Figure 1 describes ( Page 6 ) but the elements in the picture are not explained in the text ( such as ' red ' i ', ' ).
Solution : Add text description :
In Figure 1, red arrows ( ↑ ) denotes increasing, blue arrows ( ↓ ) decreases in C / N pools under drought. '
Page 5 : ' Drought reduces soil C-N ratio ' Unexplained mechanism added : ' due to disproportionate C loss via microbial mortality '
Press Page 8 : Classification of droughts ' Press vs. pulse ' Definition of a clear criteria for circular argument : ' Press : > 1 growing season ; pulse : < 3 months'
Page 13 : The future direction of ' gather additional data ' is too vague and specific : ' e.g., metatranscriptomics of rhizospheres in arid crops '
5)Other suggestions
Terminology unity :
The full text uses PGPR ( non-PCPR / PCPB ) in a unified manner.
Arbuscular Mycorrhizal Fungi ( AMF ) appeared for the first time.
Causal chain strengthening :
Page 11 : ' Drought reduces microbial activity → decreases nutrient mineralization '.
Complementary mechanism : ' by limiting substrate diffusion and inducing dormancy [ 35 ] '.
Deleting content processing :
Remove residual tags ( such as * * Deleted : * * ) to improve readability.
Final recommendations :
The logical main line is clear ( drought → soil → microorganism → plant → mitigation strategy ), but the mechanism connection needs to be strengthened.
The full text needs to be polished after grammar revision ( it is recommended to use Grammarly or professional editor ).
The chart and the text should be strictly corresponding ( supplementary Figure 1 interpretation ).
The sentence ' drought also affects the plant-soil feedback where each crop in certain region ' does not clearly explain how drought specifically affects the plant-soil feedback of crops in different regions.
It is recommended to supplement the detailed explanation of plant-soil feedback and specify how drought affects different crops or regions.
Review conclusion
Please revise carefully according to the opinions and suggestions of the reviewers. Published after revised.
Author Response
We sincerely appreciate the detailed and constructive feedback provided by the anonymous reviewers, which has significantly improved the quality of our manuscript. We are pleased to submit our revised version of the manuscript for your consideration for publication.

Reviewer 2 Report (Previous Reviewer 2)
Comments and Suggestions for Authors
The review “Impact of Drought on Soil Microbial Communities” examines the impact of drought on the soil microbiome in agricultural systems. The authors emphasize that soil microorganisms play a key role in nutrient cycling, plant health, and ecosystem stability. Drought disrupts the structure and function of microbial communities, which can lead to weakened plant–soil interactions and reduced crop productivity.
The review comprehensively summarizes the current state of knowledge on the impact of drought on the soil microbiome in agriculture.
The authors clearly identify research gaps, especially regarding long-term effects and adaptation of microorganisms, which provides directions for future research.
However, when assessing the structure of the work, I find the lack of a concluding section—a few sentences summarizing the entire review—noticeable.
Author Response
We sincerely appreciate the detailed and constructive feedback provided by the anonymous reviewers, which has significantly improved the quality of our manuscript. We are pleased to submit our revised version of the manuscript for your consideration for publication.

Reviewer 3 Report (Previous Reviewer 3)
Comments and Suggestions for Authors
Journal: Microorganisms (ISSN 2076-2607)
Manuscript ID: microorganisms-3705362
Type: Review
Title: Impact of Drought on Soil Microbial Communities
While the revised manuscript provides a coherent and well-supported presentation of the experimental findings, a few major clarifications could further strengthen its scientific rigor.
First, although the use of Aspergillus flavus is well justified, the manuscript does not explicitly state whether the strain was screened for aflatoxin production or related biosynthetic genes, a relevant safety consideration given the known toxigenic potential of this species.
Second, while descriptive statistics (means and standard deviations) are reported, the statistical methods used to determine the significance of differences between treatment groups (e.g., ANOVA or t-tests) could be more explicitly detailed, either in the methods or figure legends.
Third, although some limitations are acknowledged, a clearer discussion on how closely the experimental conditions (e.g., UV exposure, temperature regimes) reflect real-world environmental scenarios would enhance the manuscript’s applicability.
Finally, the broader implications of the results particularly the potential use of A. flavus in field-scale bioremediation could be further elaborated to contextualize the findings within applied environmental biotechnology.
The bibliography is generally strong and includes recent, relevant sources; however, it would benefit from the inclusion of landmark studies on Aspergillus flavus, aflatoxin biosynthesis, and comparative analyses with other fungal species. Strengthening the reference base with high-impact, up-to-date literature and clarifying the manuscript's conceptual contribution would improve its overall relevance and scientific merit.
Best regards.

Author Response
We sincerely appreciate the detailed and constructive feedback provided by the anonymous reviewers, which has significantly improved the quality of our manuscript. We are pleased to submit our revised version of the manuscript for your consideration for publication.

Round 2
Reviewer 3 Report (Previous Reviewer 3)
Comments and Suggestions for Authors
Journal: Microorganisms (ISSN 2076-2607)
Manuscript ID: microorganisms-3705362
Type: Review
Title: Impact of Drought on Soil Microbial Communities
The authors have addressed the previous comments and made substantial improvements to the manuscript. In its current form, the manuscript is suitable for publication.
Best regards.
This manuscript is a resubmission of an earlier submission. The following is a list of the peer review reports and author responses from that submission.
Round 1
Reviewer 1 Report
Comments and Suggestions for Authors
Manuscript Number: microorganisms- 3392728
Title: Impact of Drought on Soil Microbial Communities
This paper proposes to delete 2.Impact of drought on soil microbiomeproaches and 3.Key knowledge gaps, find out the key, and put them in the corresponding paragraphs.
The style of this paper is suggested to be written in this order, 1. Drought stress & climate change, 2. Effect of drought on soil habitat, 3. Effect of drought on soil microbial structure, 4. Effect of drought on microbiome functions. 5.Future needs and approaches.
For long content, such as 3.Effect of drought on soil microbial structure, 4.Effect of drought on microbiome functions, it is recommended to add secondary subheadings.
L74-75 The sentence is broken
Please check.。
L424 Table
Increase the header。
The language of this manuscript is too long, the logic is not clear enough, the level is not clear enough, and the language is not concise enough. Please revise.
Review conclusion
Please revise carefully according to the opinions and suggestions of the reviewers and review after the revision.

This paper proposes to delete 2.Impact of drought on soil microbiomeproaches and 3.Key knowledge gaps, find out the key, and put them in the corresponding paragraphs.
The style of this paper is suggested to be written in this order, 1. Drought stress & climate change, 2. Effect of drought on soil habitat, 3. Effect of drought on soil microbial structure, 4. Effect of drought on microbiome functions. 5.Future needs and approaches.
For long content, such as 3.Effect of drought on soil microbial structure, 4.Effect of drought on microbiome functions, it is recommended to add secondary subheadings.
The language of this manuscript is too long, the logic is not clear enough, the level is not clear enough, and the language is not concise enough. Please revise.
Reviewer 2 Report
Comments and Suggestions for Authors
Dear Authors,
Your manuscript is interesting because it shows the impact of drought on soil microbes. The layout of the article is standard for review papers. However, Authors should read the instructions on how to properly prepare a manuscript for the "Microorganisms" journal. In the text, reference numbers should be placed in square brackets [ ], and placed before the punctuation; for example [1], [1–3] or [1,3]. Prepare a document according to the guidelines for Microorganisms https://www.mdpi.com/journal/microorganisms/instructions#references
The authors did not write what the purpose of the review was. In addition, the authors could also formulate scientific (cognitive) and practical (utilitarian) goals that can be expected to be achieved as a result of the review.
The materials and methods section is missing. Even if the authors carried out a review, they should indicate the procedures they followed to achieve their objective. The authors did not write how the review presented in the article was conducted. In the literature on the subject, systematic, narrative, descriptive, critical, scoping, theoretical and other reviews are known. Since the article was classified as a review, it is worth developing the methodological aspects of the approach to conducting the review, indicating the specific method of its conduct. The manuscript does not include a detailed description of the research method, the process of data collection and analysis of the collected material, which potentially limits the readers' understanding of the background of the conducted study.
I believe that the effect of drought on enzymatic activity in soil should be described in more detail.
In my opinion, there are too few references. The content presented in many places requires confirmation by more than one article.
Figure 1 is hard to read. Improve this quality.
Good luck!
Sincerely yours
Reviewer
Reviewer 3 Report
Comments and Suggestions for Authors
Dear authors,
I have reviewed the manuscript titled "Impact of Drought on Soil Microbial Communities." It highlights how drought disrupts soil microbiomes, affecting nutrient cycling, plant health, and ecosystem stability. The study examines microbial diversity, resilience, and plant-supporting mechanisms while addressing gaps in long-term drought effects and microbial adaptations, providing valuable insights for sustainable agriculture and drought resilience.
A review of the specialized literature within the ScienceDirect Freedom Collection (Elsevier) revealed the following number of articles published: 479 in 2025, 2,854 in 2024, and 1,788 in 2023, including 2,645 review articles. In nearly all cases, the topics addressed in this research are already well-documented and extensively discussed within the existing body of literature.
What does this research add to what has already been published?
What is the degree of novelty?
Selecting and organizing bibliographic sources to include only those published within the last 2-3 years. It is a general guideline that review-type research should primarily reference scientific articles published within the last 2-3 years.
Thanks in advance for future improvements to this manuscript.
Round 2
Reviewer 2 Report
Comments and Suggestions for Authors
I accept the corrections made and recommend the manuscript for printing in its current form.
Reviewer 3 Report
Comments and Suggestions for Authors
Dear authors,
The authors have thoroughly and thoughtfully addressed the recommendations and feedback provided during the review process. Each suggestion has been carefully considered and integrated into the manuscript, resulting in a more comprehensive, scientifically rigorous, and well-structured version of their work. The revisions have significantly enhanced the clarity, depth, and overall quality of the study, ensuring that it meets the highest academic standards. Consequently, this improved manuscript is now well-prepared for publication, demonstrating the authors’ commitment to producing a valuable and impactful contribution to the field.
Congratulations to the authors.